# A Novel Approach to Engaging Communities Through the Use of Human Behaviour Change Models to Improve Companion Animal Welfare and Reduce Relinquishment

**DOI:** 10.3390/ani15071036

**Published:** 2025-04-03

**Authors:** Natalie Powdrill-Wells, Chris Bennett, Fiona Cooke, Suzanne Rogers, Jo White

**Affiliations:** 1Woodgreen Pets Charity, Kings Bush Farm, London Road, Godmanchester PE29 2NH, UK; chris.bennett@woodgreen.org.uk (C.B.); fiona.cooke@woodgreen.org.uk (F.C.); 2Human Behaviour Change for Life CIC, Norwich NR9 4DE, UK; info@hbcforlife.org (S.R.); jo@hbcforlife.org (J.W.)

**Keywords:** animal welfare, relinquishment, co-creation, community, human behaviour change, intervention design

## Abstract

By understanding the needs of the owners of vulnerable companion animals, animal welfare organisations can develop targeted community interventions with the potential to improve animal welfare and reduce relinquishment. Utilising human behaviour change models and co-creation techniques allows interventions to be designed that give consideration to the individuality of each community. This case report shares a novel approach to engaging communities towards a shared goal of promoting positive companion animal welfare.

## 1. Introduction

In the United Kingdom (UK), an estimated 60% of households have a companion animal [1]. Many of these animals are considered important members of the family and their companionship is highly valued [2,3,4]. Despite the value given to companion animals, their welfare risks being compromised due to a lack of awareness of the animals’ needs, resulting in inappropriate husbandry provision [5,6]. Examples of such compromise include animals not receiving necessary veterinary treatment [7], being housed in unsuitable environments [8], or being fed inappropriate diets [9]. Applying supportive approaches as opposed to enforcement approaches can help to improve animal welfare in the community in a scalable and sustainable fashion [10]. Supporting owners in developing knowledge and providing opportunities to learn practical skills through practising appropriate care behaviours offers significant potential for additional benefits beyond improving animal welfare including reducing companion animal relinquishment where appropriate.

Rescue centres in the UK are inundated with requests to care for companion animals who are no longer able to stay in their homes [11]. The reasons behind relinquishment are numerous and often complex [12,13,14,15,16]. Relinquishment remains an emotionally difficult decision for many owners who do not wish to give up their animals but feel they have no other choice [15]. In many cases, the decision to relinquish is made over a significant period [17]. Animal relinquishment is traditionally thought of in the context of rescue centres; however, in recent years many other relinquishment avenues, such as web-based platforms, have become available [18,19]. The traditional shelter model of intake and rehoming can be problematic. From an organisational perspective, resources are often limited and caring for animals requires significant resources in terms of time and money [20]. Despite the best efforts of rescue centre teams, the traditional shelter environment has been reported to negatively impact welfare in multiple dimensions [21,22,23]. Additional stress can also result from the separation from familiar people [24,25].

Jensen, Sandøe, and Nielsen (2020) [13] suggest that an important way to prevent relinquishment could be to work with owners to help them care for their animals. Although not all relinquishments are preventable or should be prevented, such as in the case of owner death, allergies, or in situations where welfare is significantly compromised, attempts to prevent relinquishment through interactions with owners have demonstrated a positive impact on the welfare of animals in a community [26]. Research shows that owners are open to services that can support them to keep their pets when made aware of them [17,27,28]. The types of services, identified as potentially helpful, focused on low-cost or free support such as dog training/ behaviour support, veterinary care, or pet food [28,29,30,31]. Owner awareness of the availability of services is critical [17].

Human behaviour change (HBC) models and theories can be an important element of the solution. The use of theories such as the COM-B model and Behaviour Change Wheel (BCW) [32] and the Stages of Change (SoC) from the Transtheoretical model [33] enable researchers and practitioners to develop a more detailed understanding of the key components of human behaviour and how to create behaviour change. The BCW guides the researcher through the process of understanding the factors that drive and are involved in a behaviour occurring, and it then links these to what is needed regarding intervention functions to deliver change. The SoC model provides insight into when someone might be in the process of change and therefore, acts as an aid to understanding not only why they are behaving as they are, but also what might be needed to support/facilitate change. Together, the BCW and SoC can complement each other in providing the answers to what, why, and how, namely the process to enable, facilitate, and support change to happen.

The COM-B model and BCW have been used to explore and design HBC interventions in various fields, notably human health [34,35], but also in areas related to animal welfare, such as cat confinement [36]. Similarly, the SoC model has been used to support smoking cessation [37] and encourage dietary change [38]. More recently, the model has also been applied to animal welfare topics such as reducing pet obesity [39]. One example of the successful utilisation of HBC to improve animal welfare in the community has been the Change for Animals Foundation community welfare project [40]. This project in Portugal focused on free-roaming dogs and cats and involved local community members and leaders from the beginning. Numerous other studies also support the idea that taking time to understand the problems within the community, build important relationships, and provide community members with opportunities to contribute to the project is crucial to positive animal welfare outcomes [41,42,43].

When working to prevent avoidable pet relinquishment, it is important to understand the target community [44]. Stakeholder engagement, an important element of HBC, enables researchers to build an understanding of specific community challenges and needs [45,46]. A US-based study investigating the relinquishment of cats and dogs to animal shelters for health and personal reasons identified significant regional differences in the reasons for relinquishment [47]. Reports of dog welfare concerns in Queensland, Australia, were also found to differ in areas of different socioeconomic statuses, suggesting that types of welfare problems vary between communities [48]. Exploring the attitudes and beliefs of target communities can also highlight important differences between those communities. For example, respondents in rural Spanish areas are significantly more likely than urban dwellers to view companion animals as family and agree that animals are sentient beings [3]. When looking at large dog relinquishment to municipal facilities in New York City and Washington D.C., Weiss et al. (2014) [17] found differences between the communities and suggested that consequently, different interventions would be required in different communities.

Glanville et al. [44] suggest that targeting smaller communities by exploring their unique situation is likely to be more effective than larger generic campaigns. Co-creating interventions with community members can enable the design of targeted interventions which focus on the needs of individuals within the area [49]. Co-creation can include stakeholders from different contexts such as community members, professionals, and academics [50]. Co-creation in the field of companion animal welfare is an essential element in encouraging owner engagement and ultimately improvement in animal welfare. Multiple studies investigating welfare concerns and relinquishment have found that despite owners reporting that low-cost or free support services may help them to continue to care for their animals [17,29,51,52], when such interventions are offered, owner engagement is limited [53,54]. Reasons for this are potentially numerous and complex, but by engaging communities in the development of services, co-creation can help to ensure that services are relevant and accessible to users [51]. Ensuring impact for service users is particularly important in the charity sector where resources are limited, and underuse can result in the cessation of service provisions [55].

In this paper, we present a novel approach to designing targeted interventions to help reduce companion animal relinquishment and ultimately improve animal welfare in communities using HBC models and co-creation principles. Three overarching research questions were explored throughout the project:(1)What are the challenges to pet welfare within the target community?(2)What are the pet-related support needs of pet owners in the target community?(3)Investigate a co-creation approach to the design of support services to help pet owners within the target community to positively care for their pets.

For clarification, whilst it is recognised that the term companion animal is often used within the academic literature [56], this article will use the term pet as it is accessible to both pet owners and relevant professionals. Pet also aligns with the mission of Woodgreen Pets Charity, where this project was undertaken.

## 2. Design Process

The below sections explore the steps taken during an explorative project initiated by Woodgreen Pets Charity (Cambridgeshire, England) in partnership with Human Behaviour Change for Life (HBCL) in a community in East Cambridgeshire, England. HBCL worked alongside Woodgreen in a mentorship capacity for the duration of the project.

Whilst Woodgreen had undertaken outreach activities in local communities for some time, the purposes of this project were to carry out the following:Access the most vulnerable pets and their owners at the right time.Positively impact the welfare of as many pets as possible within the community.Work towards a sustained change to pet owner behaviour to make appropriate decisions and provide a good quality of life for their pets with frontline support from within the community enabled by Woodgreen.

The project had three stages (Figure 1):

### 2.1. Understand the Community

#### 2.1.1. Desk-Based Research

Initially, time was taken to begin to build a picture of the community by exploring the existing organisational data related to the community and the internal expertise of the Woodgreen team. Organisational data including requests for support from within the target community were reviewed, and surveys were shared with Woodgreen teams, including the community team (Table 1). The surveys sought to explore the following:(1)Team member experiences of working within the target community including the pet welfare challenges encountered;(2)The pet-related needs of the community;(3)Views of wider pet-related issues.

The internal survey findings highlighted challenges with accessing the most vulnerable pets and people in the community. This became a consistent thread within the intervention development process. As a part of this, the importance of establishing partnerships within the community was raised. From their experiences, the team identified that there were challenges with staff resources in relation to being able to provide the level of support required by pet owners. Beyond accessing people requiring help, collaboration was also identified as critical to ensuring community projects could be scaled up and made sustainable. The teams raised concerns over the accessibility of veterinary care within the community and timely treatment for pets. An overarching theme was the negative impact of broader social problems on pet welfare, which confirmed the importance of considering the human element of the community.

External data sources relating to the community and national reports that focused on pet welfare were also included in the initial exploration. These sources included peer-reviewed scientific literature, governmental and non-governmental organisation publications, and communications from local partner organisations, such as local authorities and housing providers. This initial research provided a better understanding of the target community, and the team utilised these to formulate and develop the questions to be addressed in the latter stages of the project. Ideally, a full literature review would be undertaken to enhance the understanding of potential issues; however, in a charitable organisation with resource constraints and operational pressures, this will not always be possible. For this project, where operational pressures constrained the time available for a full literature review, exploration of the key literature and peer-to-peer discussions were undertaken and provided significant insight.

#### 2.1.2. Mapping

Following the initial exploration of the community data, a participatory workshop was undertaken with Woodgreen staff to facilitate an in-depth consideration of three elements of the community: locations, stakeholders, and influencers. This session was attended by team members from across the charity including community outreach, education, animal behaviour and training support, and veterinary professionals. The workshop began with a community mapping exercise focused on exploring community activity, locations where it was occurring, and who was involved (Figure 2). Following the workshop, the combined outputs of the groups were input into the mapping software MapHub (free version [57]), which could identify the specific locations within the target community in practice.

Next, the focus shifted to the stakeholders and identifying those potentially involved in interactions with pet owners in the community. Veterinary professionals were identified as key stakeholders. The value of this stakeholder analysis exercise included thinking creatively to consider pet owners who may have different accessibility requirements through engagement with sectors without a direct link to pets, such as housing and citizens’ advice [58]. The importance of collaboration as highlighted by the Woodgreen team survey meant that identifying potential partners for both the development and delivery of future programmes was important [55,59].

Social influencers have the potential to affect change behaviours within their communities and are, therefore, important to consider when designing interventions [60]. Influencer mapping was used to identify key pet-related community influencers, such as those providing pet advice on community social media groups or local cat rescue supporters. Within this session, different categories of influence were identified including professional, social, environmental, and policies and legislation.

#### 2.1.3. Community-Based Research

Community-based research was undertaken to gain a holistic understanding of the facilitators and challenges within the community and the attitudes, beliefs, and behaviours of the people who live there. Ethical approval was obtained from the HBCL Ethics board (Ref HBCL002WG). Informed consent was gathered from all the participants involved in the research and co-creation elements. The participants had to be over 18 years old and live within the target community. Debriefing information, including how to withdraw from the research process, was included at all stages.

In this phase, a mixed-methods approach was used to capture the key information whilst retaining the voice of the community [61,62]. Two surveys (Table 2), including qualitative and quantitative elements, were deployed to gain a broad understanding of pet ownership within the community and the behaviours of those who own pets when it comes to accessing veterinary services.

The decision to include a focal topic (access to veterinary services) was based on information from members of the Woodgreen team who had already undertaken outreach activity in the area. The surveys were shared through an online platform (SurveyMonkey [63]) and promoted through social media groups and partnerships within the community to try and engage harder-to-reach participants who would not typically partake in research relating to companion animal welfare [64]. Flexible, creative methods for participant recruitment including face-to-face recruitment at venues provided through partnerships within the community to help generate a representative sample [65]. A separate, complementary survey (Table 2) was shared with local professionals who interacted with pet owners, and this included veterinary professionals, pet shop owners, local authority staff, and housing association teams.

This stage of research sought to address five research questions:(1)Which types of pets are owned within the target community?(2)Where do pet owners seek advice related to their pets?(3)What is the awareness of Woodgreen and pet-related support service provision within the area?(4)What factors influence owners’ provision of veterinary care for their pets?(5)What are the broader animal welfare issues within the target community?

Based on the survey findings, six interviews with community members were undertaken. For this project, the survey-first approach was taken due to the knowledge and experience of the community team who had recently provided services in the area and to support interview participant recruitment. The interviews were semi-structured, focusing on the key aspects highlighted from the community and local professionals surveys, whilst allowing the participants to share what was important to them. The interview participants were recruited through the earlier survey and community contacts such as housing associations. Four of the interviews were undertaken online (Microsoft Teams, version 24335.208.3315.1951 [66]) and recorded for transcription and analysis. Due to the practical nature of this project, two interviews were undertaken via telephone and were unable to be recorded, and in these cases, substantial notes were taken and later digitised. This approach of recording some interviews and taking notes for others has been successfully used previously in projects within the community [4,67].

Descriptive statistics were produced for the quantitative survey data and an approach based on thematic analysis was undertaken for qualitative interview data [68]. Due to this being an operational project at a charity, resource pressures meant that a full thematic analysis was impractical. However, parts of the six-stage process were used to extract meaning from the data, including familiarisation with the transcripts, code generation and theme exploration for consideration in the later stages of the project. Where more time allows, a full thematic analysis could be undertaken.

#### 2.1.4. Community Research Findings

The survey findings highlighted a lack of awareness of the support services available from Woodgreen beyond rehoming and relinquishment. They also revealed that many community members would not go to animal welfare organisations as a source of information on pet health. Numerous concerns were raised relating to taking pets to the vet including finances, time, whether the visit was necessary, and the anxiety that some owners experienced in relation to visits. For owners of multiple pet species, decisions around the veterinary care sought were species-specific.

The findings from the survey of community professionals highlighted concerns surrounding pets not receiving timely veterinary care, with 75% of the respondents either disagreeing or strongly disagreeing with the statement “Pet owners are seeking timely treatment for pet health problems”. A total of 11 pet professionals responded to the survey, which is a significant proportion of the total population of pet professionals in the area.

Four core themes emerged from the interviews:**The importance (and challenges) of community**

Almost all the interviewees reflected on the importance of community in relation to pets. In many cases, this was viewed positively, such as enabling social interaction through dog walking groups or encouraging community integration through initiatives such as ‘dog days’ delivered by social housing providers. Conversations about pets were considered commonplace within the community and were facilitated by noticeboards and social media platforms. Social media was recognised as key within this community. Individuals would ask for support with their pets, including requests for food provision; discuss stray or missing cats; and advertise pets needing homes.

Community conversations were considered helpful by some, but considered unwelcome by others, particularly when it came to unsolicited advice. Whilst some participants deliberately avoided advice from others in the community (particularly when approached on dog walks) over concerns that the advice was often poor quality, others would go out of their way to offer advice, regardless of whether it was wanted. One participant highlighted the judgement of others when it comes to pet ownership stating, “You don’t get bad pets, you just get bad owners”. Some participants were concerned by the ease of communication relating to pets, particularly for rehoming adverts. Both pet owners and local professionals raised that some community members struggle not to overburden themselves when they see an animal needing help.


**Decision making and action planning**


The theme of decision making came through in the interviews in two different ways: Firstly, decision making over when and how to take a pet to the vet. Concerns were raised over making the decision about whether a pet needed to attend the vets, and that it is difficult when you are not aware of an issue until it is highlighted by someone else. An example of this was a participant who listed a range of potentially worrying symptoms for her elderly dog but stated she would only visit the vet when something was ‘’really wrong’’. The cost of veterinary care was raised by many as a concern, but other barriers also featured in participant decision making. One participant reflected that sometimes it was challenging because of how quickly decisions need to be made about a pet’s care, whilst another was concerned about leaving symptoms for too long in case they became worse. Worries about how pets would cope with a vet visit were common, as one respondent explained the following:

“Now the problem is if there was something wrong with Fred, to get him [sic] would be a problem because you can’t pick him up. I like my skin on my body and he’s a big strong cat.”(Participant 1)

For similar reasons, other respondents would only use a vet in an emergency for their pets. Accessibility of the vets in the local area, with particular reference to limited opening hours, was also reported as a barrier. This meant that some individuals were travelling outside of the local area for veterinary care, but not all had access to transport to facilitate this. They, therefore, had to fit visits around the local veterinary practice hours or were unable to access veterinary care.

Decision making relating to pet acquisition was also discussed. The participants reflected on this in different ways. Some had taken an organised and considered approach to acquiring their own pets, whereas others were much more emotionally driven. One participant with experience in supporting social housing tenants reported that this seemed to be much more of a problem with cats than dogs. The impulsive acquisition of cats was reflected in interviews where pet owners explained they felt that although they were never looking for pets, they just ‘’seem to find them’’. Where emotionally driven decisions relating to pet acquisition had been made, some participants also reported concerns about affording the veterinary care of multiple pets and in some cases, charitable support for this had previously been sought.


**Views of charity**


There was a general lack of awareness of the support services available from Woodgreen. Whilst Woodgreen rehoming services were recognised and trusted by the majority of the participants, only those with direct experience of the other support services reported being aware of them.

When the participants had used a Woodgreen service, they were positive about their experience. Upon being made aware of the availability of owner support services, the participants explained that they would expect a good level of service provision and that recognised standards and accreditation would be important. Some participants held the view that it was a good service but they did not feel it was for people like them, as they knew where to go for help already and were able to access services like their local vets when needed.

“I think my assumption has always been that that’s (charitable services) for people who couldn’t… who either are like I need to solve this problem, or I’ll give up my dog, or who really can’t afford to go elsewhere […].”(Participant 2)

In-person or digital engagement, via websites, with other animal welfare charities was mentioned but not common for most of the participants. Some feedback relating to charitable support services was negative. It was raised that some potential users may view charities similarly to social services, with concerns that organisations may remove their pets.


**Well meaning but misguided**


The participants painted a picture of a community of well meaning but sometimes misguided pet owners. They believed that some individuals were taking on more pets than they could realistically manage and may struggle to provide for them. One of the local professional participants said:

“I’ve got one example of a lady who we’re working with at the moment, and she’s already got 2 cats and she’s gone and got another one, but she can’t really… She can’t afford the third one, but she couldn’t resist it.”(Participant 6)

The participants felt that there were generally good intentions and positive attitudes towards pets within the community.

“People who’ve got the pets have got them because they want them, because they think a lot about them, want care for them.”(Participant 6)

Whilst intentional mistreatment towards pets in the community was not raised as a concern, there were signs that individuals were unable to provide fully for their animal’s needs. The participants reported seeing owners requesting help with their pets on local social media sites, such as when unable to afford pet food. Similarly, the participants reported trying to find workarounds for their own pets’ medical problems. In one such example, a participant had set up a homemade hydrotherapy pool in their garden for their elderly dog. The participants raised the issue of missing cats within the community, but within the same discussion often highlighted that community members, including themselves, were feeding cats they did not own.

### 2.2. Application of Human Behaviour Change Models

Once the research phases were completed and a baseline understanding of the community was reached, the next step was to define the problems identified. For this stage and future steps relating to behaviour change, worksheets based upon the Behaviour Change Wheel (BCW) framework (Figure 3) [69] were used to guide thinking and allow a detailed analysis through an HBC lens. The first stage of the BCW process focused on defining the problem in behavioural terms; identifying the behaviour(s) that needed to change, who was performing the behaviour and where the behaviour was occurring (Table 3). For this project, the definition was developed using the BCW process worksheet 1.

Multiple problems may be identified within individual communities. There may also be many potential options for human behaviour changes leading to positive outcomes for a single problem, but it is important to consider one at a time. To help understand which individual behaviour(s) the intervention design should focus on, step two of the BCW process was followed and a long list of target behaviours that could be changed for a successful outcome were identified. To prioritise these options, each behaviour was reviewed by considering the following criteria from worksheet 2 [69]:How much of an impact changing the behaviour will have on the desired outcome;How likely it is that the behaviour can be changed;How likely it is that the behaviour will have a positive or negative impact on other related behaviours;How easy it will be to measure the behaviour

Through considering these criteria, the selected target behaviour(s) was identified. Getting pet owners in the community to access an annual health check for their pets was chosen. This decision was reached as an annual health check would help to identify health problems at an early stage, which would lead to improved care/welfare. The likelihood of getting pet owners to partake in annual health checks was “promising” depending upon how they were delivered. Regular interactions with professionals undertaking pet health checks could also have a positive impact on owners approaching organisations for support with other pet-related queries and raising awareness of additional help available. Discussions surrounding the measurement of the behaviour concluded that the record-keeping of pets who attend health checks meant some impact measures would be available.

Once the target behaviour was identified, steps to clearly describe the behaviour were taken. Table 4 shows a summarised example of the description of the target behaviour including consideration of the six factors from worksheet 3 of the BCW [69].

With the target behaviour appropriately defined, steps were taken to identify what needs to change in more detail. To achieve this in a systematic fashion, the change required for the target behaviour to occur was broken down through the framework of the COM-B model. The COM-B model of human behaviour change explores the link between three constructs (capability, opportunity, and motivation) and behaviour [32], and is at the centre of the BCW process. This validated model is often used in the field of human health behaviour research [70,71] and is now beginning to feature more in research focusing on human behaviours towards animals. Examples of this include containing cats [36], reducing obesity in horses [72], donkey whipping [73], and visiting marine parks [74]. Each of the COM-B concepts is further divided into two factors as shown in Figure 3. Capability is made up of physical capability (e.g., a person’s physical function) and psychological capability (e.g., a person’s mental function). Opportunity includes both physical (e.g., financial and other physical resources) plus social opportunity (e.g., social norms and community resources). The motivation construct is a combination of reflective (e.g., conscious processes) and automatic (e.g., habits) motivation. The Theoretical Domains Framework [69] expands on each of the COM-B constructs and can be used where time allows to further consider the change required.

To help ensure that the research phase findings were thoroughly considered in this stage, the themes were mapped onto the different constructs of the COM-B model using a dual-step approach. Firstly, worksheets 4 and 4a from the BCW were used to work through each construct of the COM-B model and Theoretical Domains Framework systematically to ensure a comprehensive understanding of all the possible influences on the behaviour and to link the behaviour to evidence-based intervention design and implementation [69] (Table 5). Following this, to ensure the inclusion of all the relevant information, the COM-B constructs were mapped back onto the key research findings.

#### 2.2.1. Intervention Development

Step 5 of the BCW process supports the identification of appropriate intervention functions based on the diagnosis of the behaviour that needs to be changed from the previous stage (worksheet 4) [69]. The findings of the deeper COM-B analysis were mapped onto the intervention functions in accordance with the guide. The BCW sets out nine intervention functions to consider in relation to the COM-B components. These are education; persuasion; incentivisation; coercion; training; restriction; environmental restructuring; modelling; and enablement. Each function was reviewed in relation to the Woodgreen services available and whether they were something the charity could offer in future. Worksheet 5 from the BCW process was used to assess the intervention types relevant to the identified behaviour against the APEASE criteria of being Affordable, Practical, Effective, Acceptable, Side-effects, and Equity, ensuring that interventions designed would be appropriate [69].

The final steps of designing the initial intervention idea were completed based on elements of worksheets 6–8 from the BCW process [69]. This enabled the team to consider policy categories, behaviour change techniques, and the potential modes of intervention delivery. Considering these elements through the APPEASE lens, and in relation to the existing Woodgreen services that may be suited to improving outcomes for the target behaviour, made a good starting point for intervention development. When considering intervention types, it is important to consider the sphere of influence within which an organisation can or is currently working. For example, training and education to help improve the physical and psychological capabilities of owners when it came to organising health checks for their pets were very much a possibility. Similarly, improving the accessibility of health checks through environmental restructuring and enablement was identified as another potential opportunity.

The Stages of Change (SoC) (Transtheoretical model) was also applied to the target behaviours to identify what needs to happen, and when, for the target behaviours to occur [33]. The SoC was worked through to consider what needed to change and how the organisation could support this change based on the community-specific findings from the research phase. The SoC has five stages:Pre-contemplation—During this stage, individuals are unaware of the need to change and have no current intention to do so.Contemplation—At this stage, individuals are aware of a problem and begin to consider actions to remedy it.Preparation—Individuals in this stage are intending to take action in the near future and take steps to prepare to do so.Action—At this stage, individuals undertake an action to change their behaviour.Maintenance—During the maintenance stage, individuals have to work to prevent relapse and maintain the behaviour.

Understanding which stage a population or individual is at in relation to the target behaviour can allow tailored interventions to be developed [75]. In this project, the SoC model was used to consider how an intervention designed would be relevant and accessible to move pet owners from the pre-contemplation and contemplation stages to the preparation and action stages. The maintenance phase was also considered important in ensuring health checks were regularly sought, and this was highlighted for discussion within the co-creation stage of intervention development.

Based on the application of these two models, the project team developed the idea of offering free health checks for pets within the community. Community-based veterinary provisions have previously been suggested as an approach to addressing a lack of accessibility to veterinary services [76]. This form of intervention was considered a good option to identify potential health issues in pets at an earlier stage whilst also promoting awareness of support services the organisation could provide within the community. Given the findings from the research stage, it was considered that this form of intervention could also help to bridge the gap between pet owners and veterinary professionals to ensure more pets would be seen by a vet in a timely manner. This type of intervention was thought to offer owners a free method of accessing guidance regarding when to access vet care for their pet, something they had expressed was challenging.

#### 2.2.2. Action Planning—Implementation Intentions

In this project, we considered including the principles of action planning within the health check to help bridge the intention–behaviour gap for pet owners. As part of the health check, it was decided that if a change in owner behaviour was identified as necessary, implementation intentions would be used [77,78]. Implementation intentions are a specific form of action planning that can support behaviour change. Implementation intentions have been used successfully to change human behaviour in the areas of smoking cessation [79], physical activity promotion [80], and healthy eating [81]. In the field of equine welfare, the importance of involving the individual in the formation of action plans and linking the target behaviour to a routine existing cue, for example, picking out hooves, has been demonstrated [82]. An example of how implementation intentions were considered as likely to be used within the health check intervention is provided below:

**What** do I need to do…Register my dog with a vet**Why** do I need to do it…So that if they are ill I am able to take them quickly**How** do I need to do it…I need to phone the vet on this number…**When** will I do it…Tomorrow morning after I have dropped the children at school**Who** will help me to do it…I am able to do this myself

### 2.3. Co-Creation with the Community

Three focus group co-creation sessions were carried out to inform the development of the health check intervention. The first group involved pet owners from the target community (7 participants), the second was with local professionals (11 participants), and the third with Woodgreen team members (13 participants). The focus group framework applied the principles of co-creation [83]. The respondents in the community sessions were recruited through community networks and follow-up contact with the interview participants who had expressed interest in involvement in the co-creation element of the project. Both community-based focus groups followed a similar structure with slight nuances based on the participants. The pet owner session used group discussion with the aim of engaging the diverse group in conversation. The professional group involved both discussion and participatory workshop activities. This approach was chosen as those attending were known to be accustomed to participatory meeting settings. The research phase findings were briefly shared with the groups along with the proposed intervention idea of offering free health checks for pets within the community. The groups were asked for their views about the intervention such as location, timing, and advertising. Later in the workshops, the groups undertook an exercise to identify the key elements they thought were important to see included in a health check. Whilst the two community-based groups had slight differences in the prioritisation of aspects of the checks, the overall proposed contents were very similar. The focus groups with pet professionals from the area opened up discussions about potential locations and collaboration opportunities. The focus groups were recorded using Microsoft Teams (version 24335.208.3315.1951) [66] and later transcribed for analysis. Each project team member who attended the focus groups also collated their own reflections on the sessions.

The challenges of accessing veterinary care were further discussed within the pet owner group, particularly in relation to finances, multi-pet management, and transportation to and from veterinary appointments. Financial challenges continued to be highlighted frequently.

“I only take animals to the vet when it’s a problem… I always try to treat at home first and then take them to the vet if it doesn’t work. Because of money, because of insurance.”

“I’ve got five animals and if I take them to the vet every time they need a vet I’d be skint.”

Beyond the financial barriers, physically getting pets, in particular cats, to the vets was reported as problematic.

“To actually physically get some of the cats to the vet would cause me problems.”

This led to reflection upon the appropriate location for potential health checks. Many participants felt that home visits could lessen the burden on both humans and pets. The idea of health checks was received positively, with some participants raising the concern that there was a lack of awareness of Woodgreen’s existing support services in the area, despite some prior engagement with this particular community.

“I didn’t know that you did all the neutering and that last year, so there’s obviously a lack of getting out to the general public.”

The importance of both awareness and reassurance was raised by one participant who explained that some of her friends felt unable to take their pets to the vet because of worries that the pets would be taken from them. Another participant reflected that it needed to be clear that the services were for everybody as “if I saw free I’d think it was for people on benefits”.

The focus group included a discussion of service user preferences including location, time, and booking methods. The pet owner group had mixed views on these subjects and the project team considered testing different intervention formats. There was a desire for home visits but also a belief that drop-in clinics at community locations (e.g., community centres) would be beneficial. A flexible offering of times was suggested as the needs of individuals differed, something that led to further reflection on the limited opening hours of veterinary surgery in the community. Similarly, there was variation in the booking preferences between telephone access and online systems.

A significant benefit of the co-creation sessions was the active networking, including the sharing of contacts and potential partners. The pet owner focus group shared ideas relating to influential individuals in the area who may have access to venues or oversee communication channels. The local professional’s session identified potential partners directly within the group, with individuals such as pet shop management offering locations from which to deliver the intervention.

The sessions were also used to understand what pet owners would want to gain from a health check. The key topics identified included weight and diet advice, dental checks, behaviour advice, and guidance around insurance. The suggestion of practical support and demonstrations to learn basic health checking and husbandry procedures to undertake at home were well received. The discussion also explored information to take away from the health check session and methods of reminding owners of the need for an annual health check.

The third focus group carried out with Woodgreen team members was structured similarly to those carried out in the community, but reflected more on Woodgreen’s role in delivering this type of intervention. This included consideration of marketing materials and required collateral, for example, health check records. Once again, this co-creation approach encouraged engagement throughout the organisation.

Applying the additional contribution of the co-creation element of the design process, the final intervention idea to trial in the target community was a free pet health check—to include an action planning element when appropriate. The health check would be promoted within the community via the channels the residents had identified as important: social media and local stores. Partnerships formed through collaboration with professionals would also help with promotion, such as through the local pet shop and housing association. Based upon co-creation feedback, a combination of community-based clinics and home visits would be offered, spread across different times, and days of the week.

## 3. Discussion

Human behaviour and intervention design frameworks can effectively support the design of animal welfare-focused HBC interventions within target communities. Removing assumptions by taking the time to understand what the community challenges or needs are, and instead co-creating interventions with individuals from the area, informs the production of bespoke interventions. Applying these key principles to such projects ensures that future community programmes are effectively underpinned by theory to produce practical interventions for change.

By utilising HBC theories and models, such projects enable animal welfare problems to be understood, informing practical solutions. Many other approaches to interventions have attempted to improve animal welfare or reduce relinquishment rates, yet they often report limited success [53]. Viewing community challenges through an HBC lens reveals a complex picture. Whilst financial constraints were a concern for many community members, accessibility issues such as opening hours or animal transport issues were also key barriers to accessing veterinary care for their pets.

This project implemented a framework to address the many limitations raised by these previous intervention studies, particularly the individuality of different communities [44]. As part of their review of the relinquishment of companion animals, Protopova and Gunter [53] reported that owner commitment to an intervention remained problematic even when its financial cost was low. A study by Gunter et al. [54] provides a good example of this; despite inviting adopters and their new dogs to group dog walks with free access to a trainer, only 12.6% of the owners included in the intervention attended a walk. Similarly, Blackwell, Casey, and Bradshaw [84] found that compliance with advice provided to help reduce the risk of separation anxiety in newly adopted dogs was poor. In this project, engaging the community in the intervention development facilitated consideration of aspects that owners highlighted as being important in the accessibility of veterinary services, such as opening hours and location. Studies investigating barriers to veterinary care have highlighted the accessibility of services as important [76]. The focus group discussions highlighted that one solution does not suit all and encouraged the development of a more flexible approach including home visits. Home visits reduce access barriers for owners and transportation stress for animals. Home visits allow for more pets to be health-checked in a single home, opening opportunities for a wider variety of species and consolidating resources to see multiple pets in one location. Home visits also offer opportunities to provide advice about the animal’s home environment and care, something which is particularly important for smaller species welfare, who are less likely to be taken to the vet [85].

A key strength of the co-creation approach is that it encourages community engagement. Animal control officers in a qualitative study by Moss et al. [86] reflected on the importance of a “two-way street” in community engagement. Similarly, McDonald and Clements [87] highlighted the importance of community participation and partnerships in engaging harder-to-reach populations with respect to managing unowned cat populations. The participants in both the research and co-creation phases reflected that they would consider being potential ambassadors for Woodgreen’s community work. In their study on subsidised veterinary services in Canada, Baker et al. [88] highlight the importance of community champions in successful community engagement. An additional positive consequence of such a level of engagement within the community could be a general improvement in animal welfare in the area. Dolan, Weiss, and Slater [26] reported individuals in a community that had received spay/neuter support in the previous year from the American Society for the Prevention of Cruelty to Animals (ASPCA) were more aware of where to receive support for their companion animals and perceived that animal welfare in the area had improved overall. Similarly, Scarlett and Johnston [89] reported that calls to animal control services, and dog- and cat-related complaints, decreased following the opening of a subsidised spay/neuter clinic in North Carolina. The aim is that each interaction with the charity will lead to the transmission of information on where to seek help with pets, over time resulting in community-wide awareness of where to go for reliable, welfare-focused pet advice.

Lack of knowledge of welfare needs; inappropriate social grouping, diet, and environment; poor management of pain; and delayed euthanasia have all been highlighted as priority welfare issues for pets in the UK [5]. This means that pets may be suffering unnecessarily even when in secure homes. Drawing on discussions from the co-creation sessions for pet owners, projects designed via this framework may access pets beyond those who would never be relinquished, but are, nonetheless, experiencing compromised welfare. Many people involved in the project reported concerns about their pets but did not know where to access help. Wu et al. [90] found that people with (dis)abilities were often unaware of pet-related support available, highlighting that particular groups of potential service users may need additional support to access services, which is where this community co-creation is beneficial. Highlighting the availability of support services from charities like Woodgreen throughout the research phases of such projects will increase awareness of such support in the community prior to the intervention. Weiss et al. [17] found that when individuals are more aware of support services, such as when a helpline number is prominently displayed on a webpage, they are more likely to seek help prior to dog relinquishment, compared to areas where the service is not as well known.

Involving local professionals within both the understanding and design phases of the project highlighted the importance of partnerships and collaborations. Animal control personnel have reported these types of partnerships to be effective in improving community programmes [86]. The participants in the co-creation focus groups held the view that these partnerships were likely to be mutually beneficial, particularly for organisations that support people with potentially vulnerable pets. One such example of engagement through this project was with a representative from a social housing provider in the area. Partnerships like these can help to support individuals within social housing to care for their pets and to make informed pet acquisition decisions in the future. Housing issues are one of the most common reasons for pet relinquishment [91]; therefore, working with community organisations that are not traditionally pet-focused offers critical support to those in need. Through building awareness of the pet-related support available for their service users, professionals become an additional channel to access the most vulnerable pets and people in the community. Co-creation also enabled potential challenges to be addressed at an early stage of intervention design. For example, initially, veterinary professionals in the community showed hesitancy about WG offering free health checks, due to professional boundaries and financial impacts on business. However, the project team was able to allay anxiety and explain the intervention was designed with the aim of increasing the number of pets in the local community visiting vets. The veterinary professionals also had the opportunity to input into the structure of the health checks. Previous studies investigating low-cost spay and neuter programmes have found that such programmes do not result in substitution of the procedure source and actually increase business [92].

Beyond pet welfare-related reasons for considering the change to a preventative approach from the traditional shelter model, the impact on the humans connected with pets should also be recognised. Firstly, an approach that reduces relinquishment could save people from the emotional distress of giving up a much-loved pet [15]. Secondly, preventing the relinquishment of pets, particularly those with welfare concerns that could have been prevented through earlier intervention, could have positive psychological impacts on staff and volunteers at animal welfare charities. An example of such a process change is a change to cat management methods in Banyule, Australia [93]. Given that compassion fatigue and burnout are prevalent problems in animal care-related industries, due to the risk of continued exposure to challenging and sometimes traumatic situations, ways must be found to reduce the mental burden on those involved in animal welfare [94,95].

To ensure long-lasting impact, community projects must consider sustainability throughout their development. This is particularly important when looking to scale the project in the future, especially with finite resources. To help individuals sustain their own behaviour change, an important element of interventions designed through this process includes the demonstration and trialling of practical skills such as health-checking pets. By improving the capabilities of owners to identify signs of health concerns at an earlier stage, more pets in the community should receive earlier veterinary intervention. With limited charity resources available, service users should not become reliant on services but instead learn from them for continued change in the future. At a broader level, it is important for the community to both encourage and facilitate continued change, helping to create new social norms. This helps to ensure that services can become both sustainable at a population level and scalable. A collaborative, multi-stakeholder approach can help to provide the resources and engagement required to achieve this [96].

### Limitations and Further Considerations

The project team faced some challenges with participant recruitment through the research phases, which may present a potential limitation of the methodology. This was particularly evident in the limited responses related to pet species beyond dogs and cats. One potential limitation is the low number of pet owner interviews completed due to time constraints. At the project’s outset, this community was considered a potentially receptive audience because Woodgreen had undertaken outreach activity in the area during the previous 12 months. Despite this prior engagement, many local residents were unaware of Woodgreen’s work and this highlighted that additional challenges may be faced in areas without previous engagement. In such cases, it may be optimal to start the process at a different stage or add additional engagement opportunities/activities prior to project commencement. In areas where knowledge of the community is limited, it can be appropriate to investigate and understand salient issues within the community prior to surveying the wider population. This could be through engagement with pet owners or via interactions with the professionals supporting them.

The proposed approach has been developed with regard to the practical challenges facing many animal welfare organisations. It is, therefore, hoped that by considering the steps above, community engagement programmes based on HBC theory and frameworks can be designed to provide better outcomes for both pets and their people. As in this project, operational pressures may present limitations to undertaking all of the proposed steps; however, by considering HBC frameworks and co-creating with the community as far as practically possible, a bespoke service suited to the end-users can be achieved. The amount of time invested in the understanding stage should be maximised wherever possible to increase the likelihood of success in the latter project stages. When possible, access to someone who can guide project teams through the process of HBC or upskilling internally could be beneficial. In this project, Woodgreen worked under the guidance of HBCL who were able to provide valuable support and guidance through the various stages.

Whilst this approach is applicable to different communities, it should be noted that this project represents a single case study community. The next step for this project is to apply the approach in different communities, further testing the learnings from this project and developing a structured monitoring and evaluation programme. The mid-term aim is to develop a toolkit to guide other organisations through the practical application of the approach.

## 4. Conclusions

The framework outlined in this paper provides animal welfare organisations with guidelines for designing targeted community engagement interventions to help both pets and their people. The core elements of the approach can be followed even when project teams have limited experience in the area. By removing assumptions around the needs of the community, this process uses participatory engagement to understand the needs of the community through participatory engagement. Building upon this understanding by applying HBC frameworks means that the problem and potential solutions are viewed in a holistic manner. By involving the community, a sustainable intervention that is relevant to the community’s situation can be developed. Co-creation encourages engagement and creates ambassadors within the community, leading to innovative ideas and a better life for pets.

## Figures and Tables

**Figure 1 animals-15-01036-f001:**
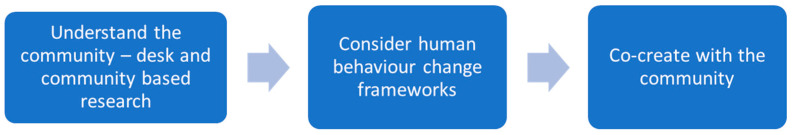
The stages of the design process undertaken during this explorative project.

**Figure 2 animals-15-01036-f002:**
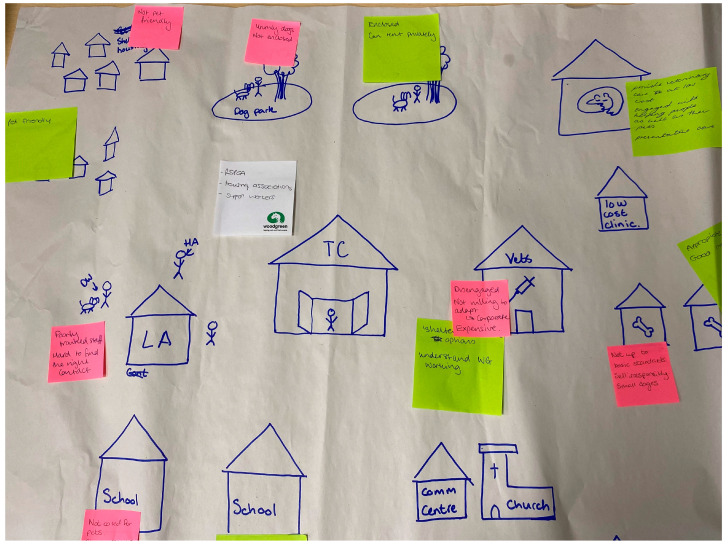
Mapping activity output where workshop attendees considered locations within the community that pet owners may frequent.

**Figure 3 animals-15-01036-f003:**
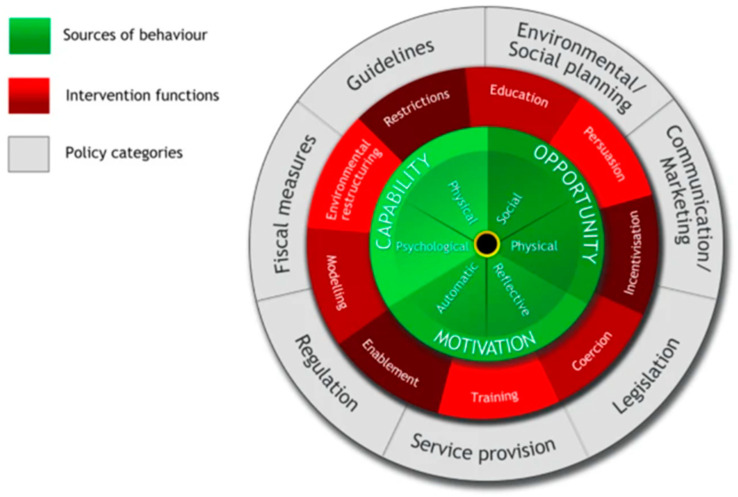
The Behaviour Change Wheel (permission granted to reproduce) [69].

**Table 1 animals-15-01036-t001:** Structure and response numbers for internal surveys.

Target Participants	Survey Themes	Responses Received
Community team members	About you and your role Exploring opportunities and challenges Future of your role Skills and knowledge needs Final thoughts	13
Wider Woodgreen team	About you and your role Exploring opportunities and challenges Future of your role Skills and knowledge needs Final thoughts	17

**Table 2 animals-15-01036-t002:** Structure and response numbers for community-based surveys.

Target Participants	Survey Themes	Responses Received
Pet owners with the Littleport community	Pet ownership Advice sources used for pet information Seeking veterinary care and recognising health problems in pets Awareness of support services for pet owners Demographic information	69
Littleport community members (including those who own pets)	Awareness of Woodgreen and services Companion animal welfare issues within the community and beyond Improving animal welfare in Littleport Pet owners only—Use of veterinary and other pet-related services These questions were designed with behaviour change principles in mind including SoC and COM-B.	56
Professionals working with pet owners in the Littleport community	Animal welfare problems within the community Sources of information used by pet owners Pet owner decision making around veterinary treatment Awareness of support services for pet owners	11

**Table 3 animals-15-01036-t003:** Example of defining the problem in behavioural terms using worksheet 1 from the BCW [69].

What behaviour?	Pet owners within the target community need to take their pets for timely veterinary treatment.
Where does the behaviour occur?	Veterinary clinic
Who is involved in performing the behaviour?	Pet owners within the target community

**Table 4 animals-15-01036-t004:** Specifying the target behaviour through the use of worksheet 3 from the BCW process [69].

1	Who needs to perform the behaviour?	Pet owners in the target community
2	What do they need to do differently to achieve the desired change?	Get a health check for their pet.
3	When do they need to do it?	Once a year
4	Where do they need to do it?	Veterinary clinic, Woodgreen clinic, home visits
5	How often do they need to do it?	Annually
6	With whom do they need to do it	Woodgreen or a vet

**Table 5 animals-15-01036-t005:** Summarised example of worksheet 4 from the BCW process [69].

COM-B Components	What Needs to Happen for the Target Behaviour to Occur?	Is There a Need to Change?
Physical capability	Owners need to be able to book and attend a health check Owner needs to be able to transport and handle animals for appointment	Maybe, for some health checks need to be more accessible
Psychological capability	Owner is aware of the health check provision and how to book Owner can remember to book and go to an appointment	Maybe, for some communication needs to be provided on multiple channels and different booking methods available
Physical opportunity	Owner has money to pay for a check Owner has time to attend an appointment Owner has access to transport to travel to an appointment with a pet	Yes, important to consider accessibility for all in the design of health check
Social opportunity	Other pet owners in the community can also be seen to be accessing health check Support from family and friends to attend the health check	Yes, we know that not all pet owners are performing this as a norm. There is a need to make the decision to attend an annual health check socially desirable.
Reflective motivation	Pet owners need to understand the benefits of preventative health Owners need to feel confident they can access the health check Owners feel health checks are an important part of being a responsible owner	Yes, there is a need for clearer communication of the welfare benefits and cost savings of preventative care
Automatic motivation	Pet owners feel positive emotional feedback and reinforcement in seeing their pet being cared for in a positive way	Maybe, some pet owners are taking on preventative health checks but not all
Behavioural diagnosis of the relevant COM-B components	The focus should be on making health checks accessible to all owners; they should be seen as normal behaviour in the community for pet owners to do and the benefits of preventative health care should be widely known and understood.	

## Data Availability

The interviews presented in this article are not readily available because the data are part of ongoing service development activity. Requests to access the interview transcripts should be directed to the corresponding author.

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
