# Peer review of "A Novel Approach to Engaging Communities Through the Use of Human Behaviour Change Models to Improve Companion Animal Welfare and Reduce Relinquishment"

_animals, 2025, doi:10.3390/ani15071036_

Round 1
Reviewer 1 Report
Comments and Suggestions for Authors
I went with interest through this paper that essentially promotes the application of long standing methodologies and tools - already used in public health education and promotion programs - to public education on responsible pet ownership and dog population management, with a focus on animal relinquishment.
Perhaps it would be advisable to include the structure of the community and local professionals surveys aiming at (1) exploring team member experiences of working within the target community and highlighting their views about the needs of the community (2) gaining a broad understanding of pet ownership within the community and the behaviours of those who own pets when it comes to accessing veterinary services.
The low number of interviews should be also highlighted in the limitations.
To be noted a misprint at Lines 157-158 , where "the purpose of this project was" should be replaced with "the purposes on this project were"
Reviewer 2 Report
Comments and Suggestions for Authors
Dear Authors,
Thank you for the opportunity to review your manuscript. I appreciate the importance and relevance of the project you are presenting. However, I have some feedback regarding the data presented, which may not align with the standards of rigorous scientific practice.
It appears that the manuscript attempts to cover too many aspects, which may distract from the essential core of the research. I believe that some elements included would be better suited as preliminary work rather than being central to the manuscript.
One of the main concerns I have is with the Materials and Methods section, which lacks sufficient detail. It is unclear who the target population was in your study. Specifically, who was invited to participate? Why was a mixed-methods approach chosen? Additionally, there is no information about the overall number of participants in both the workshops and the studies, which is crucial for transparency. This lack of qualitative and quantitative data collection makes it difficult to evaluate the findings, as there is also no results section provided.
I encourage you to reconsider how you present your data and to clarify the specific research questions driving your empirical investigations. Currently, while the project's objectives are outlined, the important goals and research questions relevant to the empirical methods seem to be missing.
Overall, I see significant potential in your project. However, as it stands, it does not fully meet the essential criteria of qualitative and quantitative data collection.
Comments on the Quality of English Language
No comment
Reviewer 3 Report
Comments and Suggestions for Authors
This manuscript presents the a novel approach to engaging communities through the use of human behaviour change models to improve companion animal welfare and reduce relinquishment. The manuscript is very interesting and provides information about problems related to the welfare of dogs and cats and the possibilities of improving them.
In the abstract, you should add the most significant results of your research so that the reader is familiar with them before reading the manuscript. The introduction contains adequate information about the problem to be examined, but it is too long. The design process is explained but some parts belong to the introduction (Line 516-522). The discussion follows the results obtained in the previous section. Conclusions derived from the results obtained.
